# Real-Time Classification of Invasive Plant Seeds Based on Improved YOLOv5 with Attention Mechanism

**Lianghai Yang [1,2], Jing Yan [1], Huiru Li [1], Xinyue Cao [1], Binjie Ge [1], Zhechen Qi [2,3,*] and Xiaoling Yan [1,*]**

[1] Eastern China Conservation Centre for Wild Endangered Plant Resources, Shanghai Chenshan Botanical Garden, Shanghai 201602, China; 15150595752@163.com (L.Y.); yan.jing01@163.com (J.Y.); wwwlihuiru@163.com (H.L.); caoxinyue0417@163.com (X.C.); gebinjie@csnbgsh.cn (B.G.)

[2] Zhejiang Province Key Laboratory of Plant Secondary Metabolism and Regulation, College of Life Sciences and Medicine, Zhejiang Sci-Tech University, Hangzhou 310018, China

[3] Shaoxing Academy of Biomedicine, Zhejiang Sci-Tech University, Shaoxing 312366, China

\* Correspondence: zqi@zstu.edu.cn (Z.Q.); xlyan@cemps.ac.cn (X.Y.)

**Abstract:** Seeds of exotic plants transported with imported goods pose a risk of alien species invasion in cross-border transportation and logistics. It is critical to develop stringent inspection and quarantine protocols with active management to control the import and export accompanied by exotic seeds. As a result, a method for promptly identifying exotic plant seeds is urgently needed. In this study, we built a database containing 3000 images of seeds of 12 invasive plants and proposed an improved YOLOv5 target detection algorithm that incorporates a channel attention mechanism. Given that certain seeds in the same family and genus are very similar in appearance and are thus difficult to differentiate, we improved the size model of the initial anchor box to converge better; moreover, we introduce three attention modules, SENet, CBAM, and ECA-Net, to enhance the extraction of globally important features while suppressing the weakening of irrelevant features, thereby effectively solving the problem of automated inspection of similar species. Experiments on an invasive alien plant seed data set reveal that the improved network model fused with ECA-Net requires only a small increase in parameters when compared to the original YOLOv5 network model and achieved greater classification and detection accuracy without affecting detection speed.

**Keywords:** weed seeds; seed identification; target detection; convolutional neural network; YOLOv5



## 1. Introduction

Invasion by alien species refers to the introduction of certain organisms into a new ecological environment through natural or man-made activities from a different place of origin, which then reproduce and spread independently in the new environment and eventually exhibit a significant ecological impact, endangering the local biodiversity [1]. China is among the countries with the richest biodiversity in the world. The impact of alien species in China has become increasingly severe with the increased frequency of worldwide commerce and transit of products and persons, as well as the rapid expansion of international tourism and logistics industries. As of 2020, more than 660 invasive alien species have been discovered in China, with 370 of them being invasive plants [2]. Every year, invasive species cause massive environmental and economic damages totaling more than USD \$30 billion.

In terms of biosecurity protocols, customs inspection and quarantine are considered first passes of defense against alien species invasion and the most critical part of invasive alien species prevention and control management. In 2020, the Chinese customs intercepted 69,500 batches of 384 species of quarantine pests and 4270 batches of 1258 alien species among inbound and outbound articles. As the world's largest grain importer, strict inspection and quarantine of imported grain are critical to the biosecurity of China. Taking Huangpu and Nansha ports in Guangdong province as an example, more than 4 million

tons of sorghum imported from the United States alone were detected with a maximum quantity of 100 seeds per kilogram of 106 species of invasive plant seeds in 19 families from 2014 to 2016 [3]. The interception frequency and content of invasive plant seeds are high, and the huge quantity has brought great difficulties to the detection of customs staff. The rapid and accurate detection of the seeds of some notorious invasive plants is related to the biological security and food security of our country. Because some seeds in the same genus exhibit only slight differences in morphological characteristics, customs staff cannot successfully identify them without the assistance of specialists. Moreover, the traditional artificial detection and identification procedure is difficult, time-consuming, and inefficient. Therefore, it is critical to establish an invasive plant seed image data set and an automatic classification system.

With the rapid development of computer vision technology, the detection method based on image recognition technology of the species has been widely adopted, the secondary detection, identification, and classification of various species have achieved significant effect. The image-based target detection task involves identifying the target objects in the image, detecting their location, and determining their category. The target detection algorithm is divided into the traditional artificial feature extraction algorithm and the convolutional neural network (CNN) feature extraction algorithm. In the early stage, traditional machine learning methods were mainly used. Chtioui et al. [4] extracted the size, shape, and texture features of the four color seed images, respectively, and then used the stepwise discriminant analysis method and artificial neural network as the classifier to classify the four seeds and achieved the recognition rates of 92% and 99%. However, due to the small experimental data set and the prior knowledge required to extract features, this result has no practical generalization ability. The traditional feature extraction methods rely heavily on features designed based on prior knowledge. If the extracted features are insufficient, it will seriously affect the accuracy of recognition.

Thanks to the quick advances of GPU hardware and parallel computing performance, CNN (convolutional neural network) represented by AlexNet [5] algorithm provides new approaches for image recognition. In addition, the subsequent R-CNN (region with CNN features) [6] algorithm was widely adopted in image recognition due to its significantly improved performance of deep learning methods in target detection tasks. CNN-based deep learning algorithms are mainly divided into two categories: One is a two-stage target detection algorithm based on R-CNN and its improved algorithms, such as Fast R-CNN [7], Faster R-CNN [8], Mask R-CNN [9], etc. The other is a one-stage target detection algorithm, such as the SSD (Single Shot MultiBox Detector) [10] algorithm and the YOLO (You Only Look Once) [11–14] series algorithm. Compared with the two-stage target detection algorithm, one-stage does not need to map the anchor box to the feature map but directly classifies and regresses the anchor box. Overall, the classification detection accuracy of the one-stage target detection algorithm will be slightly lower than that of the two-stage target detection algorithm, but the detection speed will be enhanced. CNN-based deep learning algorithms have also been widely used in seed image recognition. Javanmardi et al. [15] employ a deep CNN as a general feature extractor for corn seed varieties. Luo et al. [16] compared the classification and detection capabilities of six popular CNN-based deep learning algorithms (AlexNet, VGG-16, Xception, GoogleNet, SqueezeNet, NasNet-Mobile) for weed seeds. Loddo et al. [17] proposed a new deep convolutional neural network, SeedNet, and compared it with other convolutional neural networks in two plant seed data sets. SeedNet can achieve the best results in terms of comprehensive performance. However, as a lightweight target detection algorithm, YOLOv5 has a faster classification and detection speed. Kundu et al. [18] developed a "Mixed Cropping Seed Classifier and Quality Tester (MCSCQT)" system using the YOLOv5 algorithm, which can classify the healthy and diseased seeds of pearl millet and maize. However, the application of the YOLOv5 algorithm in weed seed is scarce so far.

Therefore, we adopted the YOLOv5 algorithm to classify and detect invasive plant seeds in real time in this paper. The YOLO algorithm was proposed in 2016 and has under-

gone development from v1 to v5. YOLOv5 is a deep learning model based on PyTorch [19]. Benefiting from PyTorch's mature ecosystem, YOLOv5's environment support and model deployment are simpler and easier. YOLOv5 algorithm belongs to the one-stage target detection algorithm. Compared with the two-stage target detection algorithm, YOLOv5 has a smaller weight file and faster reasoning speed of the model [19]; meanwhile, it can still surpass many two-stage target detection algorithms in public data sets to obtain higher accuracy. In the present work, we apply YOLOv5 to classify invasive plant seeds in real time by refining the size model of the initial anchor box using a produced data set comprising captured seed images of 12 invasive plant species. In addition, we propose an improved YOLOv5 target detection algorithm, which combines the ECA attention mechanism. In addition, we also compared the improved model with the original YOLOv5 algorithm and several other attention mechanisms. The improved YOLOv5 method enhances the extraction ability of key features of plant seeds and achieves greater classification and detection accuracy without compromising detection speed.

## 2. Materials and Methods

### 2.1. YOLOv5

The YOLOv5 algorithm includes four network models: YOLOv5s, YOLOv5m, YOLOv5l, and YOLOv5x, listed in order of increasing network depth and weight file size. To realize high performance on real-time classification on handheld devices, such as cell phones and tablets, we chose the YOLOv5s model for experimental training from the perspective of minimizing computational cost and network weighting in this study. The improved YOLOv5s network model architecture incorporating the ECA-Net attention mechanism is shown in Figure 1. The entire network structure is divided into four parts: input, backbone, neck, and output. A part of the unit structure of the network model is shown in Figure 2.

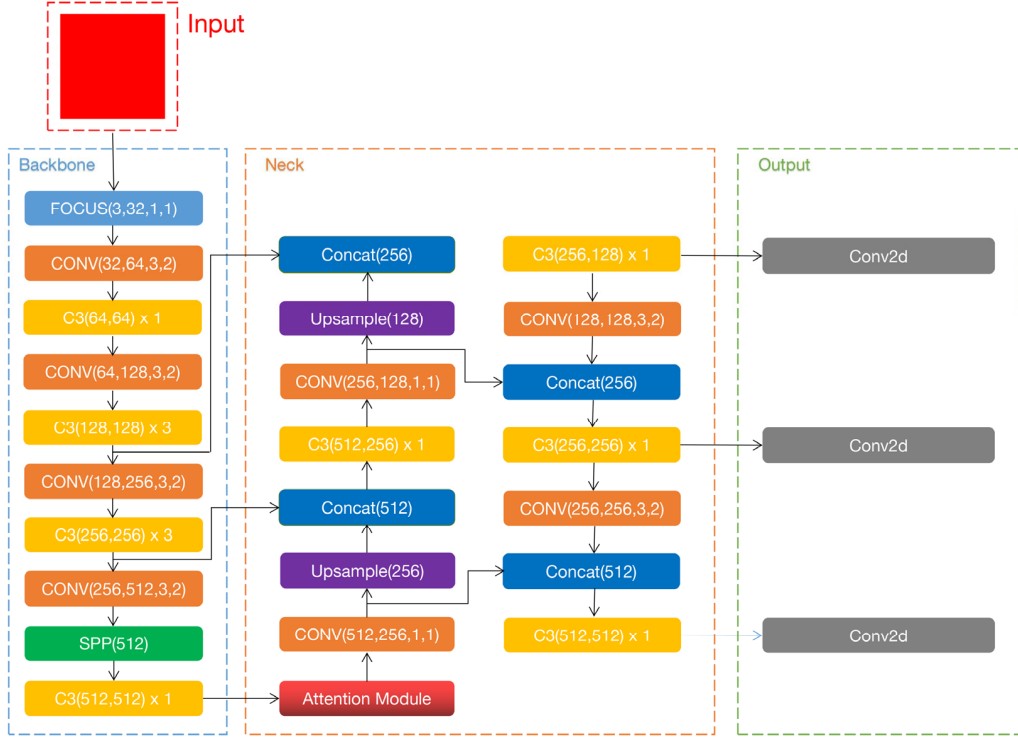

**Figure 1.** Improved structure of YOLOv5s network model. An attention module is added to the backbone part to capture the important feature information in the backbone and introduce the feature fusion layer.

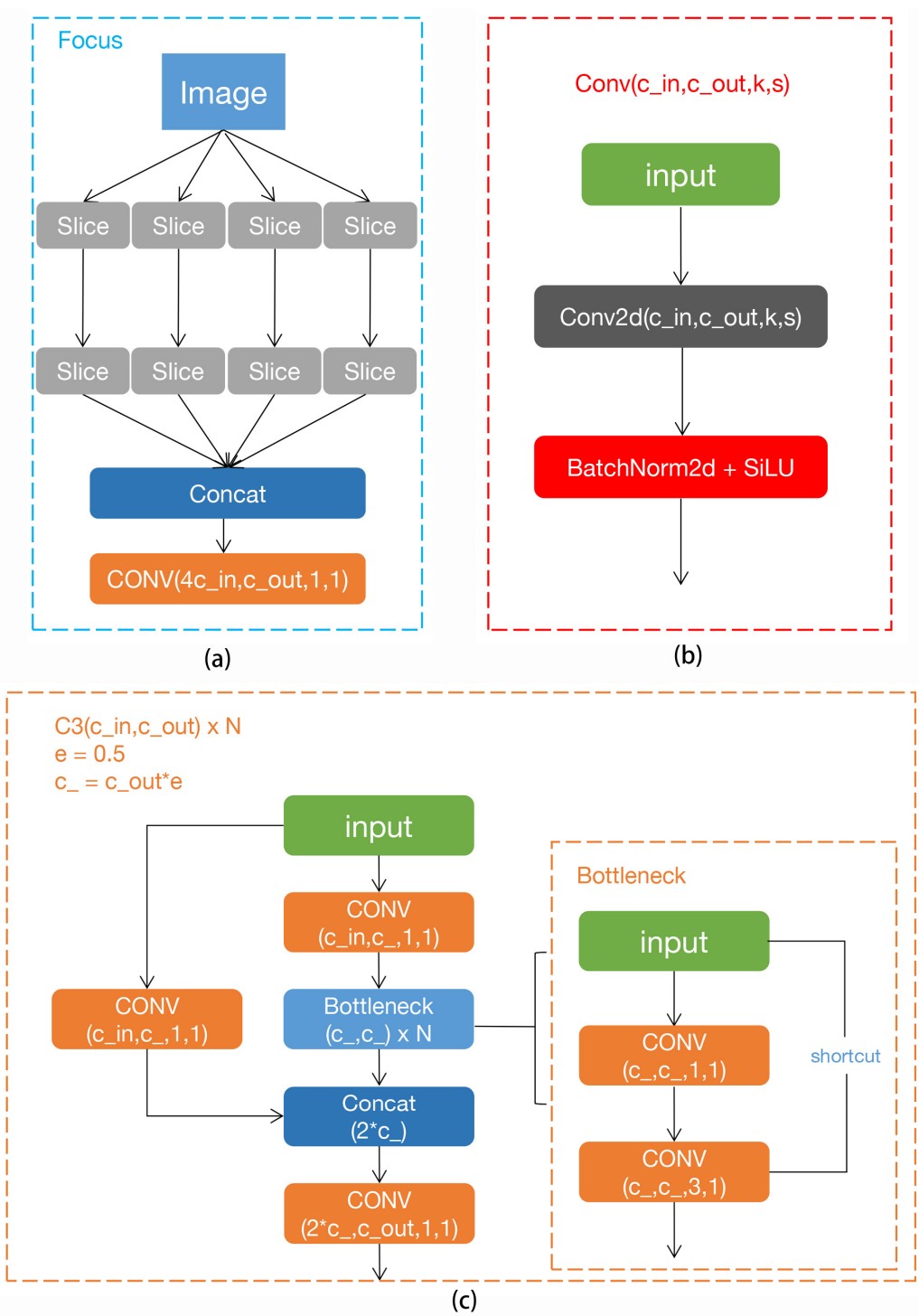

**Figure 2.** Partial cell structure in network mode: (**a**) structure of the Focus module; (**b**) structure of the Conv module; (**c**) structure of C3 module.

### 2.1.1. Input

There were fewer small targets in our self-constructed invasive plant seed data set owing to the capture of images with a macro lens, so we used the Mosaic [19] data enhancement method for the input part. Four images were scaled randomly and spliced into a picture to enhance the model's ability to detect small targets. The anchor box of YOLOv5 clustered out nine different sizes of a priori boxes under the three sizes of receptive fields based on the COCO [20] data set. Anchor boxes are statistics from all ground truth boxes in the training set, which are the most frequently occurring box shapes and sizes

in the training set. The anchor box can effectively restrict the range of predicted objects in the training process and accelerate the convergence of the model. According to our data set, we use a genetic algorithm plus *k*-means [21] to re-cluster anchors, and the new anchor box is given as follows. [87, 88, 107, 107, 116, 137], [132, 136, 152, 155, 175, 197], [230, 244, 262, 266, 352, 372]. The training phase learns the offset parameters continuously along with the zoom ratio parameters through the anchor box and ground truth. The prediction phase uses these parameters to fine-tune the anchor box, and finally, a better prediction box is obtained from the test data.

2.1.2. Backbone

The backbone of the network architecture includes Focus, C3, Conv, SPP, and ECA attention modules.

Figure 2a shows the structure of the Focus module. Its main function is the slicing operation. As shown in Figure 3, the $4 \times 4 \times 3$ image is sliced into a $2 \times 2 \times 12$ feature map. After the slicing operation of the Focus module, the double down sampling feature map without information loss is obtained. The main purpose of the Focus layer is to reduce layers, reduce parameters, reduce FLOPS (floating-point operations per second), reduce CUDA memory, increase forward and backward speed while minimally impact on model accuracy [19].

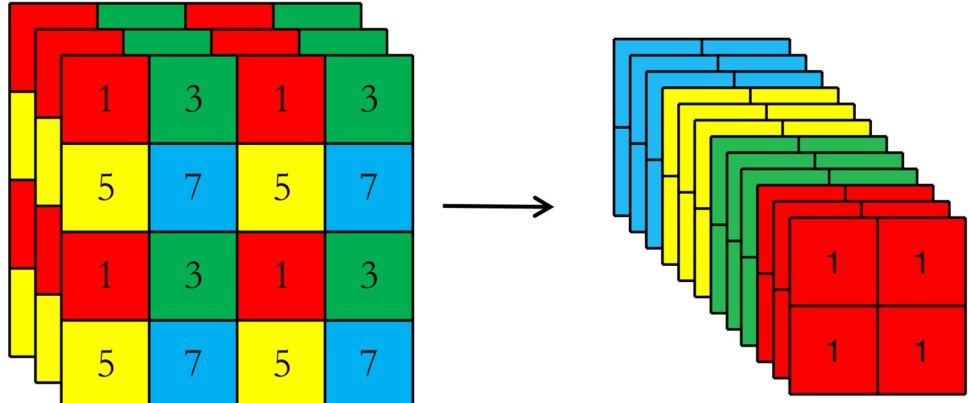

**Figure 3.** Slicing operation. $4 \times 4 \times 3$ image sliced into $2 \times 2 \times 12$ feature map.

The Conv module is a standard convolution module. A structure diagram is shown in Figure 2b. It consists of the Conv2d+BatchNorm2d+SiLU activation function. The SiLU (sigmoid linear unit) [22] activation function is used in the Conv module to further improve the detection accuracy of the algorithm and optimize the convergence effect of the model. SiLU is an improved, smoother version of the ReLU (rectified linear unit) [23] function. The formula of its activation function is as follows, and x represents the input of convolution.

$$\text{ReLU} = \max(0, x) \tag{1}$$

$$\text{Sigmoid}(x) = \frac{1}{1 + e^{-x}} \tag{2}$$

$$\text{SiLU} = x * \text{Sigmoid}(x) \tag{3}$$

The activation functions of SiLU and ReLU are similar in appearance, as shown in Figure 4. When the input value is less than 0, the output of the ReLU is always zero, as is the first-order derivative. As a result, some neurons may not be activated, and thus the corresponding parameters cannot be updated. The first-order derivative formulas of SiLU and ReLU are as follows.

$$\text{ReLU}' = \max(0, 1) \tag{4}$$

$$\text{SiLU}' = \text{Sigmoid}(x) + \text{SiLU} * (1 - \text{Sigmoid}(x)) \tag{5}$$

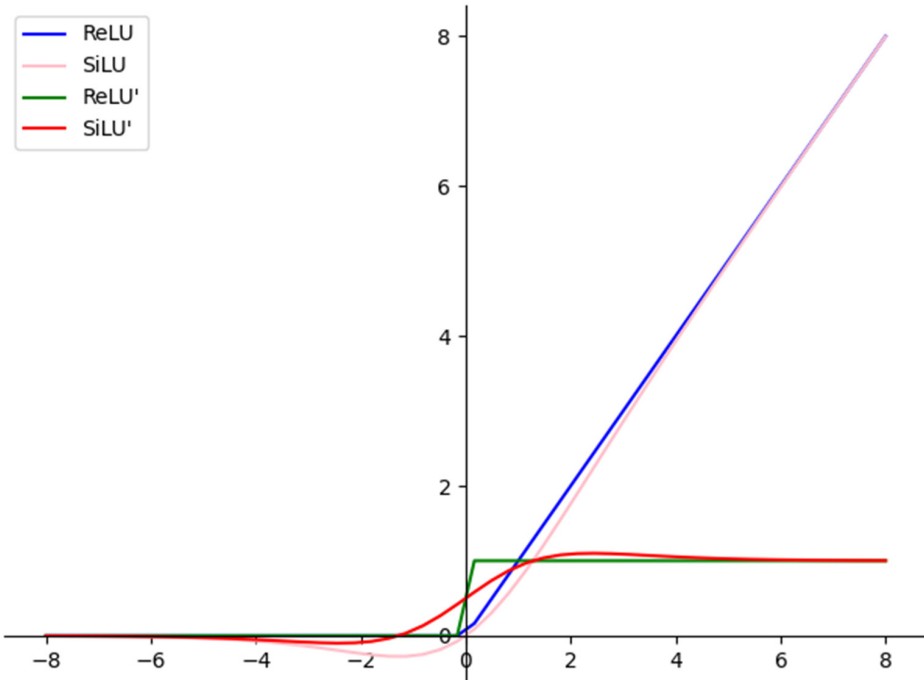

**Figure 4.** Activation functions and first derivative curves of ReLU and SiLU.

As shown by the curve marked ReLU′ in Figure 4, the ReLU activation function is not monotonically increasing, and the smooth ReLU has better stability, effectively avoiding neuron death and gradient explosion problems in the deep convolutional network.

The C3 module is a residual block based on cross-stage partial (CSP) networks and contains three convolution modules. As shown in Figure 2c, it is composed of a standard bottleneck residual module with three convolution modules. It draws on the design idea of the cross-stage local area network CSPNet [24], which can enhance the feature extraction ability and reduce the model parameters to reduce the memory cost.

As shown in Figure 5, in the spatial pyramid pooling (SPP) module [25], the feature output of different receptive fields is achieved by using three max-pooling layers with different kernel sizes. This can effectively solve the problem that the input image size of the convolutional neural network must be fixed and can avoid repeated extraction of image features to reduce computational costs.

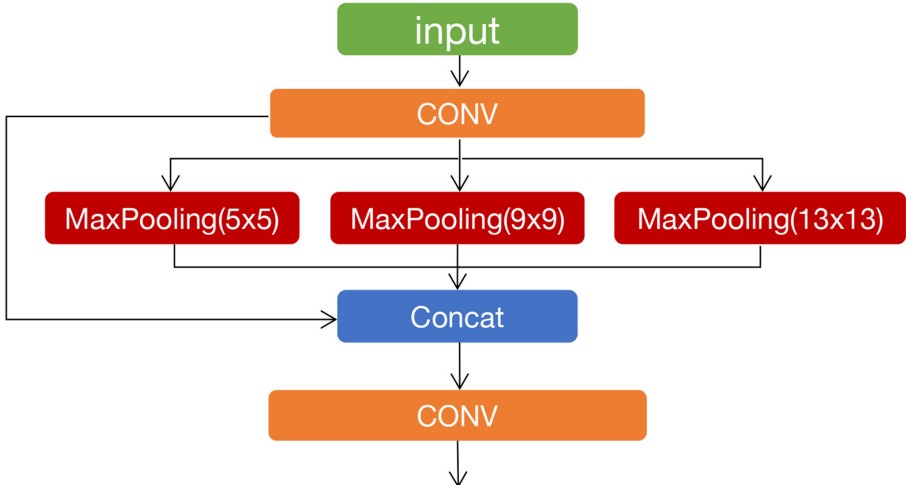

**Figure 5.** Spatial pyramid pooling structure.

### 2.1.3. Neck

The neck part adopts a structure combining an FPN (feature pyramid network) and a PAN (path augmentation network) framework, as shown in Figure 6. FPN transfers strong semantic features from top to bottom and then transfers strong positioning features upward through the bottom-up feature pyramid structure of the two PAN structures, and multiple feature fusions achieve full extraction of the three feature layers of the network.

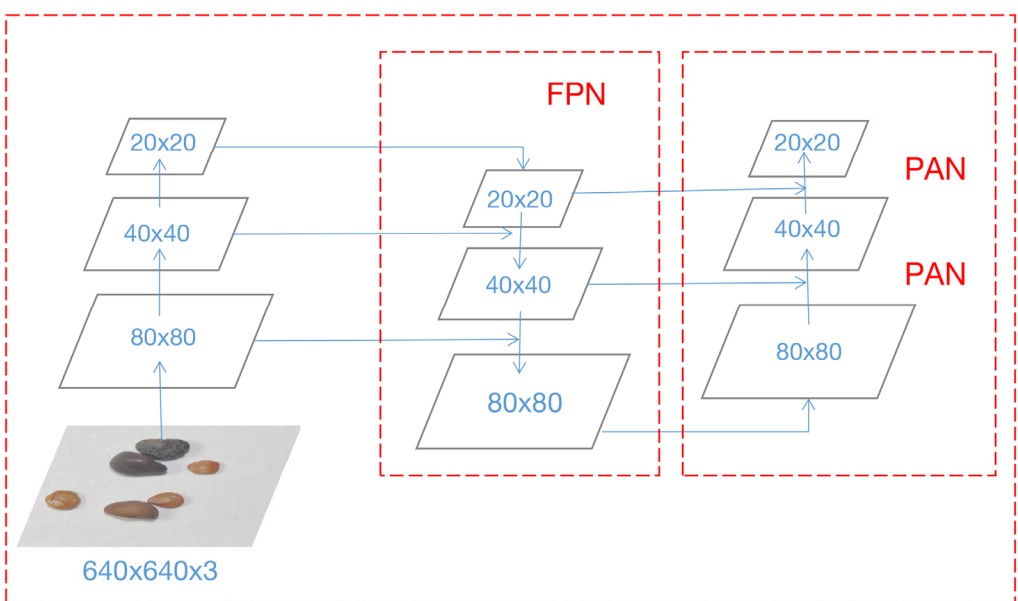

**Figure 6.** FPN+PAN structure.

### 2.1.4. Output

The output part uses GIoU_Loss (generalized intersection over union loss) function [26] for the bounding box. Compared with IoU_Loss (intersection over union loss), it increases the measurement of the intersection scale, which effectively alleviates the poor recognition caused by seed stacking in photos of invasive plants. GIoU finds a minimum closed box C to include them based on any two target boxes, A and B. It then calculates the ratio of the area of C that is not covered by A and B to the total area of C and then subtracts this ratio from the IoU of A and B, which is calculated as follows.

$$IoU = \left| \frac{A \cap B}{A \cup B} \right| \tag{6}$$

$$GIoU = IoU - \frac{|C(A \cup B)|}{C} \tag{7}$$

### 2.2. Attention Mechanism

The concept of attention mechanisms was derived from the study of human vision. Hence, attention mechanisms in deep learning are essentially similar to the selective attention mechanism of humans. Their core purpose is to select more critical information on the current task goal from many information centers while ignoring other relatively unimportant information. Attention mechanisms are mainly divided into three types, including spatial, channel, and mixed mechanisms.

### 2.2.1. Spatial Attention Mechanism

The spatial attention mechanism only focuses on an area related to the task target and only looks for the most important part of the network for processing. A representative model is the spatial transformer network (STN) [27] proposed by Google DeepMind, and

the network structure is shown in Figure 7. It contains three parts: a localization network, a grid generator, and a sampler.

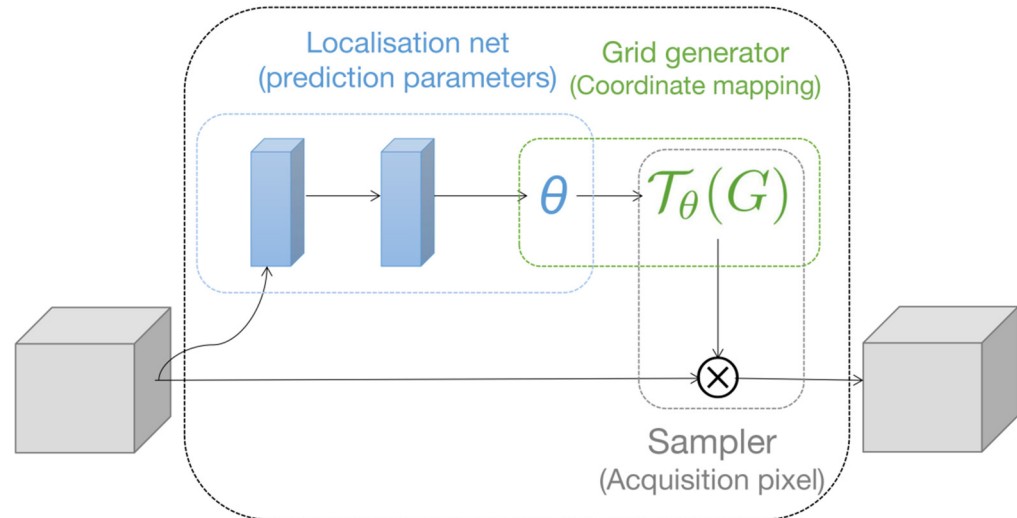

**Figure 7.** Spatial transformer network structure.

### 2.2.2. Channel Attention Mechanism

The channel attention mechanism learns the importance of each feature channel and then enhances the useful feature channels for different tasks and suppresses those that are less useful to realize the adaptive calibration of the feature channels.

As a representative model of the channel attention mechanism, Squeeze and Excitation Networks (SENet) [28] was the champion model of the 2017 ImageNet classification competition. The SE module proposed by SENet is shown in Figure 8, which mainly includes two operations: squeeze and excitation. The SE module first performs a squeeze operation on the feature map obtained by convolution to obtain the channel-level global features and then performs an excitation operation on the global features to obtain the weights of different channels and the relationship between the channels.

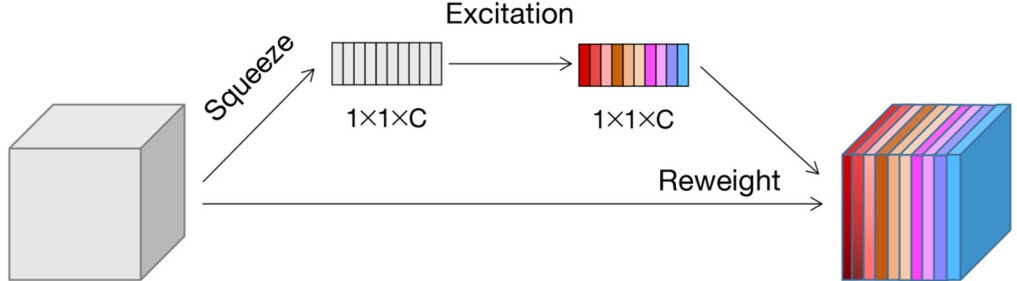

**Figure 8.** Squeeze and excitation structure.

Efficient channel attention networks (ECA-Net) [29] are introduced as an improved network based on SENet, which uses one-dimensional convolution to replace the bottleneck structure composed of two fully connected layers in SENet. The structure of the ECA module is illustrated in Figure 9. Given the aggregated features obtained by global average pooling (GAP), the local cross-channel interaction strategy without dimensionality reduction and the method of adaptively selecting the size of the convolution kernel is proposed to significantly reduce the complexity of the model while maintaining performance.

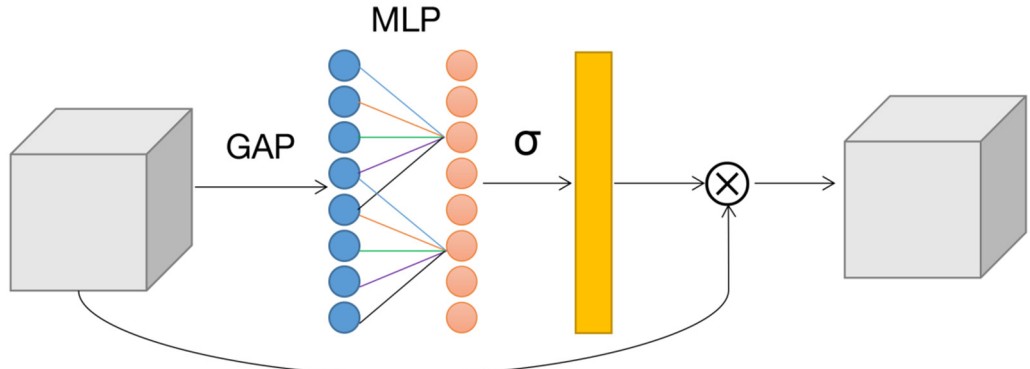

**Figure 9.** Efficient channel attention networks structure.

### 2.2.3. Mixed Attention Mechanism

The convolutional block attention module (CBAM) [30] was proposed to combine spatial and channel attention mechanisms. This model proposed a channel attention module and spatial attention module, as shown in Figure 10. The channel attention module processes the feature maps of different channels, and the spatial attention module processes the feature regions of the feature maps.

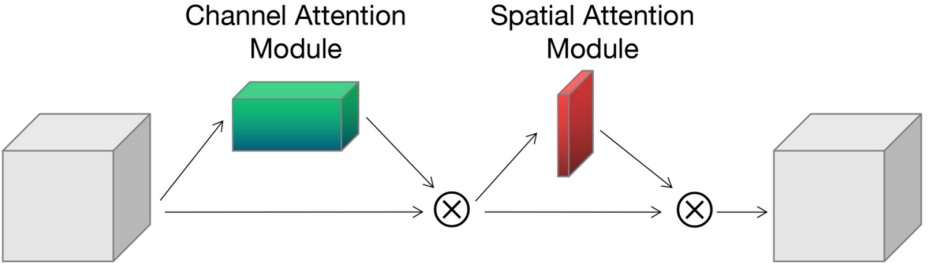

**Figure 10.** Convolutional block attention module structure.

### 3. Experiment

The hardware and software configuration of the experimental environment is shown in Table 1.

**Table 1.** Experimental environment configuration.

| Item | Configuration |
|---|---|
| OS | CentOS 7.9 |
| GPU | Tesla K80 (24GB) |
| CPU | Intel(R) Xeon(R) CPU E5-2690 v2@3.00GHz |
| Framework | Pytorch 1.7.1 |
| Data annotation | LabelImg |
| Visualization | Tensorboard |

### 3.1. Invasive Plant Seed Data Set

We selected 12 species of invasive plants in 10 genera and 7 families, and each species collected 50–80 seeds for shooting. To improve the reliability of the experiment content and results and to better fit the actual situation of customs inspection, we considered both the size of the seeds (the smallest seed of *Nicandra physalodes* (L.) Gaertner is less than 2 mm in average length and the largest seed of *Leucaena leucocephala* (Lam.) de Wit is more than 8 mm in average length) and their similarity (some sets of seeding having high similarity, such as *Solanum viarum* Dunal and *Solanum elaeagnifolium* Cav., as well as *Ipomoea lacunosa* L. and *Ipomoea triloba* L.) when selecting species to construct the invasive plant seeds data set. We did not choose seeds with a length of less than 1 mm, because too small seeds cannot

better display surface features when shooting with mobile phones. Most of the selected seeds are between 3 and 5 mm in size, and only the length of Leucaena leucocephala is 8.5 mm. The number of invasive plant seeds larger than 5 mm intercepted by the customs is small, and it is easier to identify with the naked eye due to the large individual. The details of 12 species are shown in Table 2. Figure 11 shows the seed screenshot samples from the data set, which will be closer to the shooting in the actual environment.

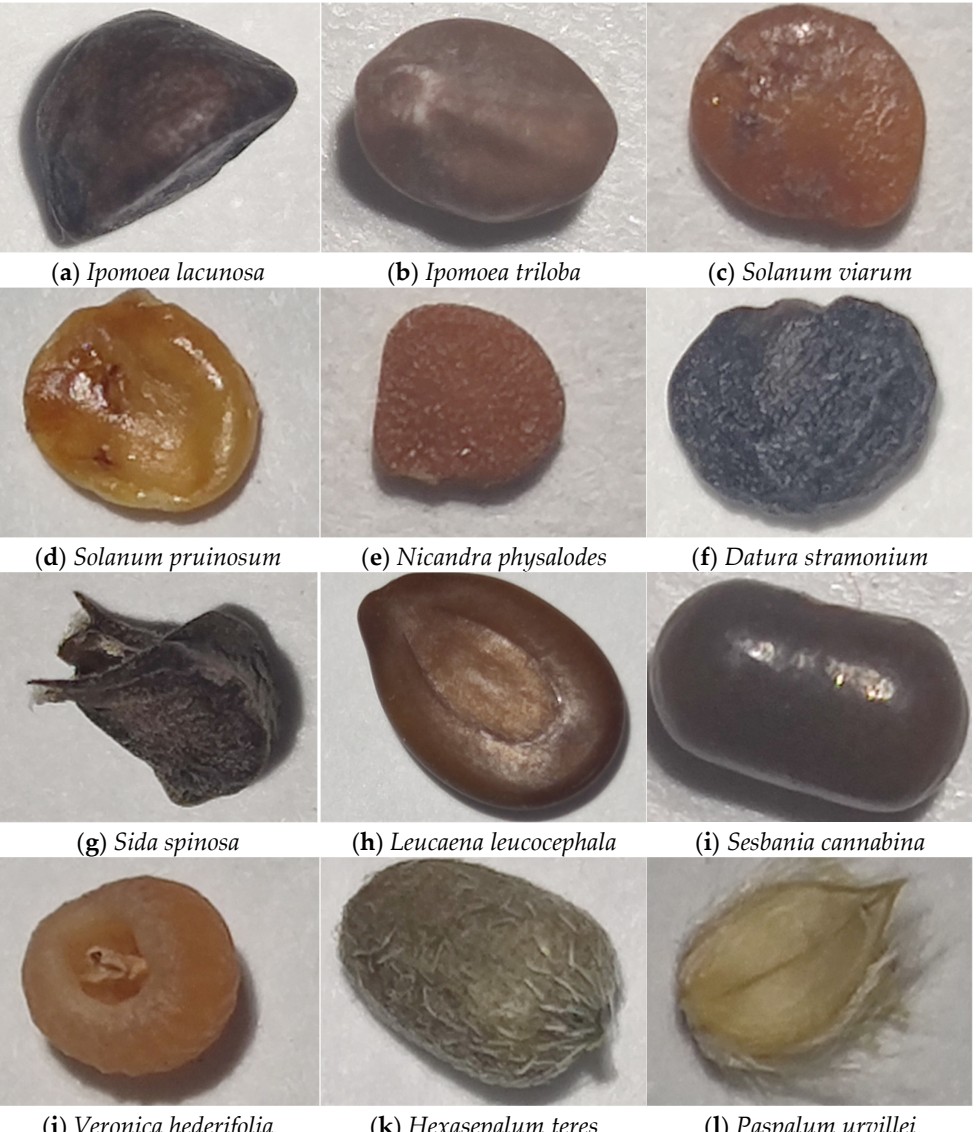

(**a**) *Ipomoea lacunosa*     (**b**) *Ipomoea triloba*     (**c**) *Solanum viarum*

(**d**) *Solanum pruinosum*     (**e**) *Nicandra physalodes*     (**f**) *Datura stramonium*

(**g**) *Sida spinosa*     (**h**) *Leucaena leucocephala*     (**i**) *Sesbania cannabina*

(**j**) *Veronica hederifolia*     (**k**) *Hexasepalum teres*     (**l**) *Paspalum urvillei*

**Figure 11.** Screenshots of 12 species in the data set.

When shooting the data set for the selected 12 species, we use the mobile phone macro lens (xiaomi10; 2 million pixels; F/2.4 aperture) to shoot the seeds under the light of 5500 K color temperature, which is more suitable for the application scenario of real-time quarantine of customs staff. To obtain more detailed features of seed appearance and improve the generalization ability of the data set, we used 5× optical zoom and 10× optical zoom to shoot seeds, respectively, taking 480 photos of single species and 520 photos of mixed species. Each picture contains a different number of targets, and some data sets are shown in Figure 12. Through random vertical and horizontal mirror flipping and random brightness adjustment, we expand the number of samples in the data set to three times the original image, a total of 3000 images and 14,682 targets. The target distribution of the 12 species is shown in Figure 13, and the target number of each species was evenly

distributed. We used the LabelImg [31] data annotation tool to label the seed data, divide the data set into a training set, and test set according to a ratio of 4:1, and the picture resolution was 4344 × 3940.

**Table 2.** Details of 12 species in the data set: including family and genus information of the species and average data of seed length, width, and height of the species.

| Scientific Name | Family | Genus | Length (mm) | Width (mm) | Height (mm) |
|---|---|---|---|---|---|
| *Ipomoea lacunosa* L. | Convolvulaceae | *Ipomoea* | 4 ± 0.3 | 3.5 ± 0.3 | 2.9 ± 0.4 |
| *Ipomoea triloba* L. | Convolvulaceae | *Ipomoea* | 4.1 ± 0.5 | 3.1 ± 0.3 | 2.5 ± 0.3 |
| *Solanum viarum* Dunal | Solanaceae | *Solanum* | 2.3 ± 0.1 | 2 ± 0.1 | 0.7 ± 0.1 |
| *Solanum pruinosum* Dunal | Solanaceae | *Solanum* | 3 ± 0.2 | 2.3 ± 0.2 | 0.9 ± 0.1 |
| *Nicandra physalodes* (L.) Gaertner | Solanaceae | *Nicandra* | 1.7 ± 0.2 | 1.6 ± 0.1 | 0.6 ± 0.1 |
| *Datura stramonium* L. | Solanaceae | *Datura* | 3.5 ± 0.2 | 2.8 ± 0.1 | 1.4 ± 0.1 |
| *Sida spinosa* L. | Malvaceae | *Sida* | 2.2 ± 0.2 | 2 ± 0.2 | 1.5 ± 0.1 |
| *Leucaena leucocephala* (Lam.) de Wit | Fabaceae | *Leucaena* | 8.5 ± 0.5 | 5.4 ± 0.4 | 1.5 ± 0.2 |
| *Sesbania cannabina* (Retz.) Poir. | Fabaceae | *Sesbania* | 4 ± 0.5 | 2.4 ± 0.5 | 1.6 ± 0.2 |
| *Veronica hederifolia* L. | Plantaginaceae | *Veronica* | 2.2 ± 0.3 | 2 ± 0.3 | 1.6 ± 0.3 |
| *Hexasepalum teres* (Walter) J. H. Kirkbr. | Rubiaceae | *Diodia* | 3.4 ± 0.2 | 2.2 ± 0.2 | 1.6 ± 0.1 |
| *Paspalum urvillei* Steud. | Poaceae | *Paspalum* | 1.8 ± 0.3 | 1.3 ± 0.2 | 0.5 ± 0.1 |

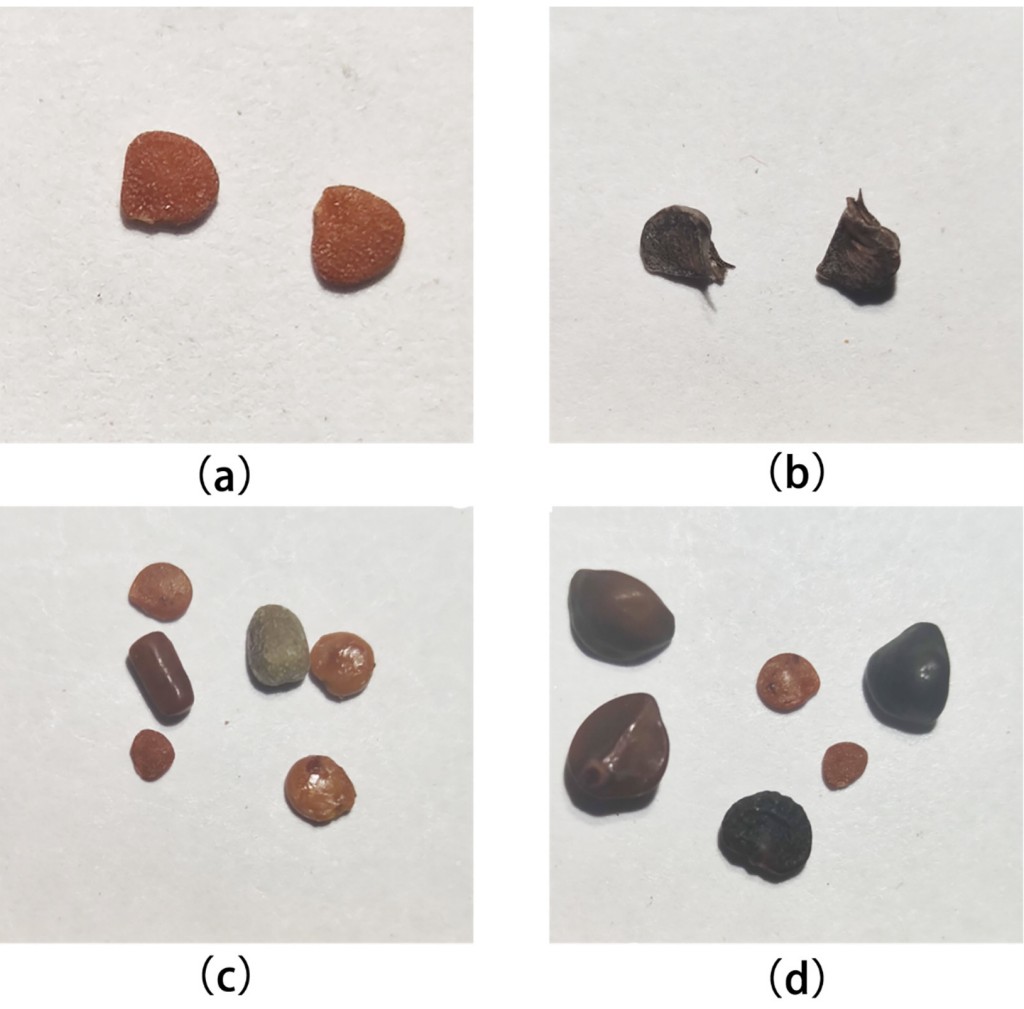

**Figure 12.** Partial data set display: (**a**,**b**) single species images in data set; (**c**,**d**) mixed species images in data set.

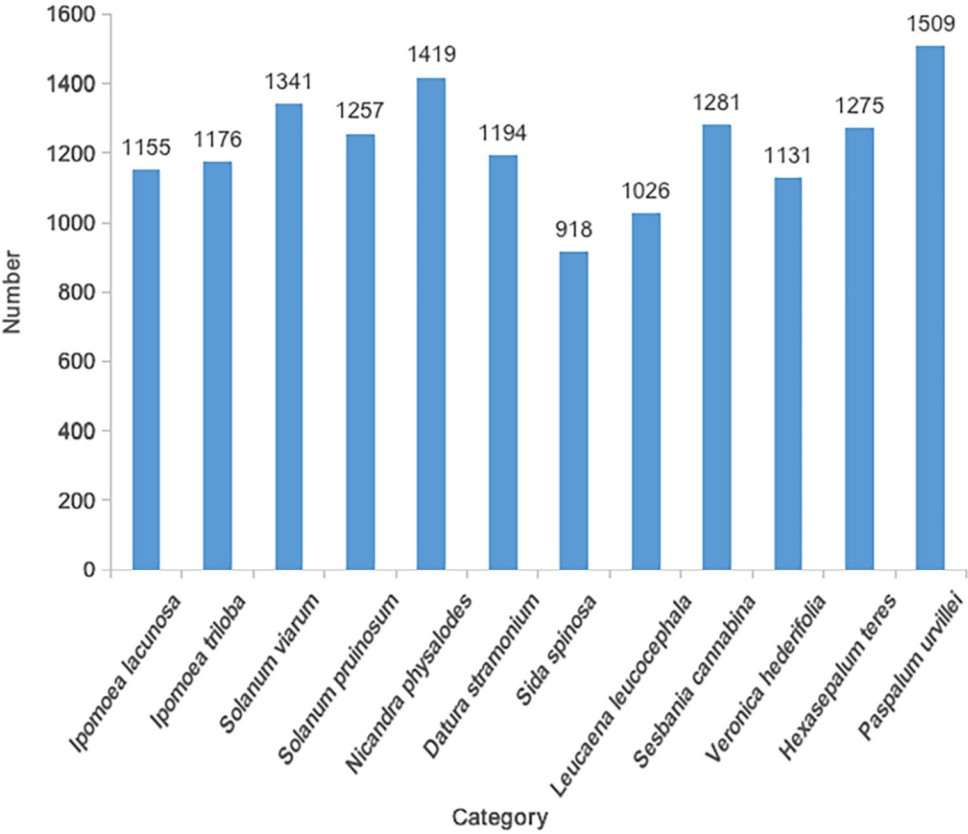

**Figure 13.** Target number statistics of each category in data set, the number of occurrences of the target seed in the data set.

*3.2. Network Performance Evaluation*

Common evaluation indicators for two-class problems are precision, recall, F1-score, mAP (mean average precision), FPS (frames per second), etc. A correctly identified sample is called a true positive (TP), whereas a negative sample incorrectly identified as a positive sample is called a false positive (FP), and a positive sample incorrectly identified as a negative sample is called a false negative (FN). Precision, recall, and F1-score are defined below.

$$\text{Precision} = \frac{\text{TP}}{\text{TP} + \text{FP}} \tag{8}$$

$$\text{Recall} = \frac{\text{FP}}{\text{FP} + \text{FN}} \tag{9}$$

$$\text{F1} - \text{score} = \frac{2 * \text{Precision} * \text{Recall}}{\text{Precision} + \text{Recall}} \tag{10}$$

According to the calculation formula of the evaluation index of the two-class problem, we derive the precision and recall rate of the multi-class problem. As shown in Table 3, we considered five classes (class 1–5) as an example (listing the details of class 1). In addition to the statistics of the five categories of predictions, we also show statistics that had content but did not predict the results and those that had no content but had the predicted results. The calculation formulas of Precision1 (P1), Recall1 (R1), and F1-score1 of class 1 are as follows.

$$\text{Precision1} = \frac{\text{TP1}}{\text{TP1} + \text{FP2} + \text{FP3} + \text{FP4} + \text{FP5} + \text{FP6}} \tag{11}$$

$$\text{Recall1} = \frac{\text{TP1}}{\text{TP1} + \text{FN2} + \text{FN3} + \text{FN4} + \text{FN5} + \text{FN6}} \tag{12}$$

$$F1 - score1 = \frac{2 * Precision1 * Recall1}{Precision1 + Recall1} \tag{13}$$

**Table 3.** Examples of five-classification precision and recall. TP (true positive), FP (false positive), and FN (false negative). In particular, when the real target is class1, but the prediction result is empty, we are defined as FN6. When there is no real target, but the prediction result is class1, we define it as FP6.

| Multi-Class | | Prediction | | | | | |
|---|---|---|---|---|---|---|---|
| | | **Class1** | **Class2** | **Class3** | **Class4** | **Class5** | **Null** |
| | class1 | TP1 | FN2 | FN3 | FN4 | FN5 | FN6 |
| | class2 | FP2 | TP2 | | | | |
| | class3 | FP3 | | TP3 | | | |
| **Real** | class4 | FP4 | | | TP4 | | |
| | class5 | FP5 | | | | TP5 | |
| | Null | FP6 | | | | | |

AP is the area under the P-R curve, indicating the accuracy in a category. The map represents the average accuracy of all categories and is used to measure the performance of the deep learning model in all categories. Among mAP@.5 indicates that the IoU (intersection over union) threshold of NMS (non-maximum suppression) is greater than the map value of 0.5, mAP@.5:.95 means that the IOU threshold is calculated every 0.05 from 0.5 to 0.95, and finally, the average value is calculated. AP and mAP are calculated as follows:

$$AP = \int_0^1 P(R) \, dR \tag{14}$$

$$mAP = \frac{\sum_{i=1}^{N} AP}{N} \tag{15}$$

where P is precision, R is recall, and N represents the number of classes in the data set.

Finally, the evaluation result was the average of all the classes. FPS represents the number of image frames that can be processed by the target detection method per second, which verifies the real-time performance of the detection method.

### 3.3. Experimental Implementations and Settings

This research integrates the SENet, CBAM, and ECA-Net modules in the backbone of YOLOv5s. We also conducted comparative experiments with the original YOLOv5s. The hyperparameter settings in the model are listed in Table 4. We refer to the application of the YOLOv5 target detection algorithm in other fields [32–34], and the network input size was 640 × 580, the batch size was set to 64 according to the performance of the GPU, and the initial learning rate was set to 0.01. The amount of data was not large; therefore, we used the Adam optimization algorithm to optimize the network parameters.

**Table 4.** Hyperparameter settings applied in the YOLOv5s.

| Hyperparameter | Image Size | Batch Size | Epoch | Optimizer | Learning Rate | Beta1 | Beta2 |
|---|---|---|---|---|---|---|---|
| Value/Type | 640 | 64 | 400 | Adam | 0.01 | 0.937 | 0.999 |

### 3.4. Experimental Results and Analysis

We tested YOLOv5s and three improved models that integrate attention mechanisms on the invasive alien plant seed data set. We recorded the differences between the models on the five aspects of Params, Precision, Recall, F1-score, and FPS. In terms of model

parameters and detection speed, the original YOLOv5s has the best effect, but after integrating the ECA attention module, the three performance evaluation indicators of the model, Precision, Recall, F1-score, mAP@.5, and mAP@.5:.95 have been improved. The results are shown in Table 5, and in Figure 14, we show the prediction results of some test sets in the YOLOv5s+ECA model.

**Table 5.** Experimental comparison of YOLOv5s integrated attention module. Bold data represent the best result.

| Models | Params | Precision/% | Recall/% | F1-Score/% | mAP@.5 | mAP @.5:.95 | FPS |
|--------|--------|-------------|----------|------------|--------|-------------|-----|
| YOLOv5s | **7,093,209** | 93.02 | 89.28 | 91.06 | 90.65 | 80.40 | **32** |
| YOLOv5s+SE | 7,414,233 | 93.09 | 88.87 | 90.99 | 90.08 | 80.52 | 28 |
| YOLOv5s+CBAM | 7,137,121 | 92.83 | 89.52 | 91.22 | 90.83 | 81.16 | 29 |
| YOLOv5s+ECA | 7,283,164 | **93.96** | **90.11** | **91.94** | **91.67** | **82.77** | 29 |

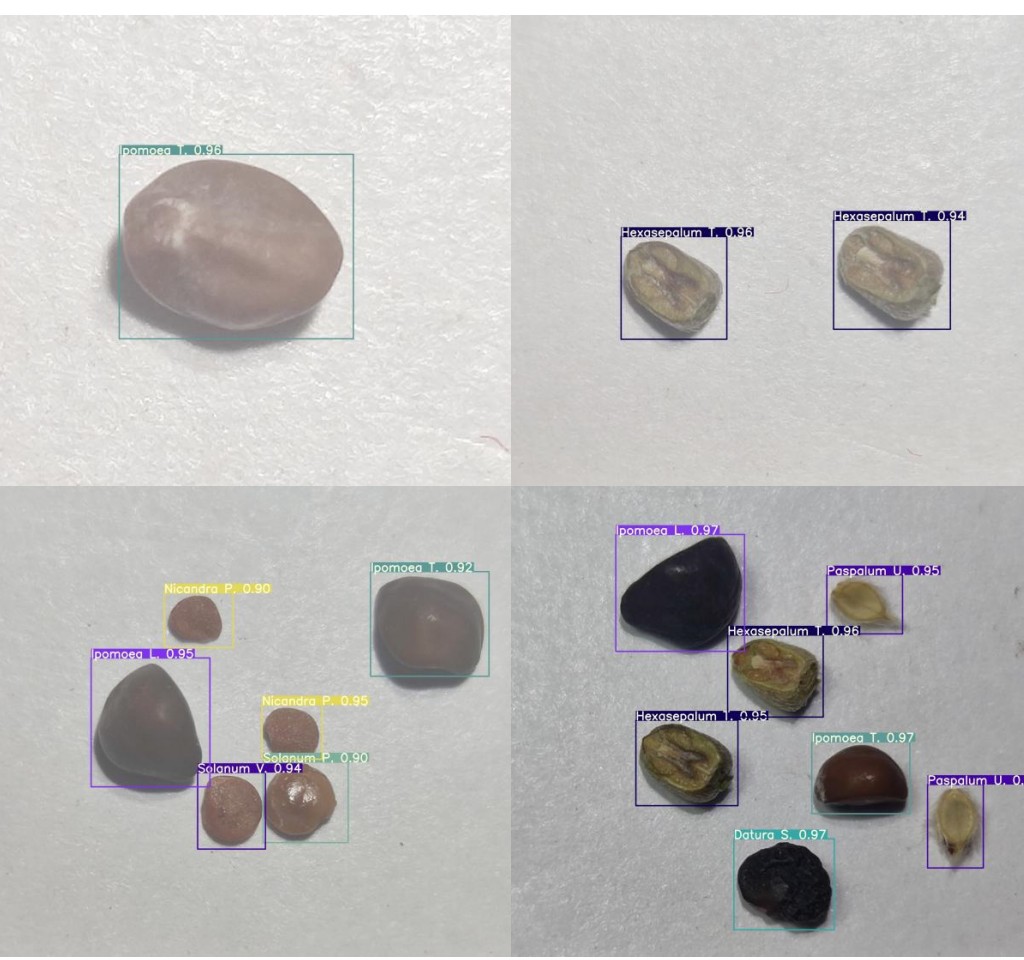

**Figure 14.** Partial prediction results of YOLOv5s+ECA model.

Data analysis shows that in terms of detection speed when the inference size was 4344, the four network models were able to achieve millisecond-level real-time classification detection using an Nvidia Tesla K80 GPU. The number of parameters of all models that incorporate the attention module has different increases in parameters. Among them, the YOLOv5+SE model has a maximum increase of 4.53% in parameters, and FPS decreased by four frames, the final F1-score has a slight decrease. The minimum increment of the YOLOv5+CBAM model parameters is only 0.62%, and FPS is reduced by three frames, but it obtains a better detection effect than YOLOv5s+SE. The final F1-score is increased by

0.16% compared to YOLOv5s. YOLOv5s+ECA model has the best optimization results. Compared with YOLOv5s, the parameters of YOLOv5s+ECA increased by 2.68%, and FPS was only reduced by three frames, but the precision, recall, F1-score, mAP@.5, and mAP@.5:.95 increased by 0.94%, 0.83%, 0.88%, 1.02%, and 2.37%, respectively.

For a single category, the test results of the four algorithms on the alien invasive plant seed data set are shown in Table 6. Compared with other individuals, Sida spinosa showed obvious differences. The F1-scores of Sida spinosa in the four models were not high. We speculate that this is because the number of samples in the data set Sida spinosa is relatively small compared to other species. Ipomoea lacunosa and Ipomoea triloba belong to the same genus and are very similar in appearance, which may lead to poor recognition of the YOLOv5s network model. However, F1 scores were improved to different degrees after fusing the SENet, CBAM, and ECA-Net modules, and the ECA-Net module had the best effect. *Sesbania cannabina* also performs poorly in the models of YOLOv5s and integrated SENet. Because the seed surface of *Sesbania cannabina* is extremely smooth to reflect light and appear light spots in the case of flash photography. We cannot effectively solve this problem by adjusting the photographing method, which may lead to some models being unable to obtain more surface features and poor performance. The YOLOv5s+ECA model exhibited suitable detection results in 12 categories and achieved the highest F1-score among the four models in six categories.

**Table 6.** Comparison of F1-score detection results for 12 invasive plant seeds based on four algorithms with the produced data sets. Bold data represent the best result.

| Category | YOLOv5s | YOLOv5s+SE | YOLOv5s+CBAM | YOLOv5s+ECA |
|---|---|---|---|---|
| *Ipomoea lacunosa* | 0.8904 | 0.8906 | 0.8976 | **0.9127** |
| *Ipomoea triloba* | 0.9056 | 0.9106 | **0.9214** | 0.9186 |
| *Solanum viarum* | **0.9440** | 0.9383 | 0.9398 | 0.9405 |
| *Solanum pruinosum* | 0.9170 | 0.9134 | 0.9087 | **0.9322** |
| *Nicandra physalodes* | 0.9071 | 0.9164 | **0.9180** | 0.9164 |
| *Datura stramonium* | 0.9153 | **0.9188** | 0.9090 | 0.9168 |
| *Sida spinosa* | 0.8921 | **0.8985** | 0.8966 | 0.8980 |
| *Leucaena leucocephala* | 0.9136 | 0.9028 | 0.9108 | **0.9182** |
| *Sesbania cannabina* | 0.8978 | 0.8890 | 0.9030 | **0.9076** |
| *Veronica hederifolia* | 0.9284 | 0.9258 | 0.9207 | **0.9406** |
| *Hexasepalum teres* | 0.9000 | 0.8934 | 0.9000 | **0.9120** |
| *Paspalum urvillei* | 0.9156 | **0.9214** | 0.9217 | 0.9187 |

Based on the above analysis, we find that the integration of YOLOv5s with the ECA-Net attention module may be expected to increase the number of model parameters only slightly and not to reduce detection speed while improving detection accuracy. Moreover, it can achieve better performance in the real-time detection of invasive plant seeds.

## 4. Discussion

### 4.1. Potential Applications

In this paper, we establish a weed data set and offer a method for automatic detection of invasive plant seeds, which has a promising potential to assist customs staff in analyzing inbound alien plant seeds. We evaluate different improved methods on the invasive plant seed data set. The results show that the detection speed of the YOLOv5 method fused with ECA-Net is close to 30 FPS, which can meet the basic requirements of real-time detection of inbound plant seeds by customs staff. However, it should be noted that many species have small differences due to the particularity of plant seeds. Therefore, in the process of real-time classification and detection of plant seeds, it is necessary to achieve sufficient illumination intensity and high enough shooting resolution so that the deep learning model can effectively read the subtle characteristics of seeds.

*4.2. Hyperparameter Exploration*

In the previous experimental implementations and settings part, we used Adam optimizer to train the model. The image input size is resized to 640, each batch of model input was 64, the initial learning rate is 0.01, and the loss stabilized after 400 training iterations. To verify whether the settings of these hyperparameters are optimal, we observe the performance changes of the YOLOv5s model fused with ECA-Net on the invasive plant seed data set by adjusting these hyperparameters. The experimental results are shown in Table 7. The six experiments of "exp1–exp6" modified different hyperparameters, and the results showed that the F1-score of exp1 was the best. Because our invasive plant seed data set is a small data set, the Adam optimizer is obviously more suitable for our data set. This also verifies that our hyperparameter settings are optimal. The F1-score of exp4 is the worst. We speculate that the learning rate is too high, which makes the depth model unable to converge effectively. An appropriate learning rate is very important for the deep learning model.

**Table 7.** Hyperparameter adjustment and results. Exp1–6 represents six experiments performed.

| Number | Image Size | Batch Size | Learning Rate | Optimizer | Epoch | F1-Score |
|--------|-----------|-----------|---------------|-----------|-------|----------|
| Exp1 | 640 | 64 | 0.01 | Adam | 400 | 91.94% |
| Exp2 | 320 | 64 | 0.01 | Adam | 400 | 90.54% |
| Exp3 | 640 | 32 | 0.01 | Adam | 400 | 91.26% |
| Exp4 | 640 | 64 | 0.1 | Adam | 400 | 71.23% |
| Exp5 | 640 | 64 | 0.001 | Adam | 400 | 91.10% |
| Exp6 | 640 | 64 | 0.01 | SGD | 400 | 90.92% |

## 5. Conclusions

Because of the difficulty of conventional customs biosecurity protocols based on human labor in detecting invasive plant seeds and the long detection process required, in this study, we proposed a real-time classification and detection network based on YOLOv5s fusion attention modules and constructed an image data set consisting of 12 types of invasive plant seeds. In order to better extract the important characteristics of invasive plant seeds and strengthen the comparison of similar species in the model, we combined the YOLOv5s network with the three attention modules of SENet, CBAM, and ECA-Net, and then conducted a comparative experiment with the original YOLOv5s network on the invasive plant seed data set. Using the invasive plant seed data set, the network model composed of the YOLOv5s fusion ECA-Net module achieved higher classification detection accuracy while only adding a small number of parameters and without loss of detection speed. By fusing the ECA-Net attention module, the ability to extract important features of similar species is enhanced, and the classification and detection accuracy of similar species are improved. In the experimental environment, the model FPS integrated with the ECA network can reach 32 frames, which can meet the real-time detection conditions of customs staff. The overall experimental results show that the modified model can achieve better results in the actual application of invasive plant seed detection.

In future studies, expanding the species and target number of the data set is crucial to improve the classification impact and practicability of the model on invasive plant seeds. Furthermore, we will continue to investigate the strategy of fusing the characteristic information of other modalities to improve the seed classification and detection accuracy, especially for the seeds of closely related species with high morphological similarity.

**Author Contributions:** Conceptualization, X.Y. and Z.Q.; methodology, L.Y. and X.C.; investigation, J.Y. and H.L.; resources, B.G.; writing—original draft preparation, L.Y.; writing—review and editing, X.Y. and Z.Q.; project administration, X.Y. and Z.Q. All authors have read and agreed to the published version of the manuscript.

**Funding:** This research was funded by the Special Fund for Scientific Research of Shanghai Landscaping & City Appearance Administrative Bureau, grant numbers G212405, G222403; National Wild Plant Germplasm Resource Center, grant number ZWGX1902; Special Fundamental Work of the Ministry of Science and Technology, grant number 2014FY120400; the Natural Science Foundation of Zhejiang Province, grant number LY21C030008; and the Open Fund of Shaoxing Academy of Biomedicine of Zhejiang Sci-Tech University, grant number SXAB202020.

**Institutional Review Board Statement:** Not applicable.

**Informed Consent Statement:** Not applicable.

**Data Availability Statement:** Not applicable.

**Acknowledgments:** We are grateful to the reviewers for their thorough reviews and suggestions that helped to improve this paper.

**Conflicts of Interest:** The authors declare no conflict of interest.

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
