# Peer review of "Real-Time Classification of Invasive Plant Seeds Based on Improved YOLOv5 with Attention Mechanism"

_diversity, doi:10.3390/d14040254_

Round 1

Reviewer 1 Report

The manuscript introduced novel deep learning based methods to conduct real-time classification of invasive plant seeds. The methods showed the efficiency in terms of high prediction accuracies. The manuscript is well written. So I will suggest an acceptance to the Diversity journal.

As a minor comment, would suggest to add a full name for the YOLOv5 algorithm. 

Reviewer 2 Report

The authors propose a functional tool for the recognition of potentially invasive seeds. For this purpose they propose the implementation of a deep learning algorithm based on the evolution of the well-known yolo algorithm.

There are many works on seed recognition and it would have been interesting to carry out a more extensive bibliographical work on the state of the art. Of course the authors present the applicative interest of their proposal but there are many works on seed recognition and practically none are cited in this work. It would be desirable that a state of the art be included in the publication by citing the oldest works (not using deep learning) to the most recent (using deep learning). Furthermore, the authors are content to use only version 5 of the Yolo algorithm, which is still controversial and not published scientifically. It is therefore imperative to compare these results with other deep learning architectures (V3, V4, R-CNN for example)

in details:

2-materials and methods

Table 1 : you must replace this table by samples of images of seeds

the input part 2,2,1 is not clear.

what is the interest of mosaic step?

you must explain the clustering step and the link with anchors box. Why is it an important step

All this part correspond to automatic processing from YoloV5 algorithm.

the Backbone part in 2,2,2:

you may explain the interest of the slicing operation (figure3), why this process improve de feature extraction results.

the dataset presentation 3,1 must be presented before in order to present image and problems comming from this kind of images.

You must present image acquisition system better. what type of images? (grayscale?, colour?, multispectral?, etc...). you may present an image from each species.

if i understand your native resolution is 4344x3940 and your resize for input deep architecture is 608x608 (information only in fig 6), so how does it work on very small seeds wich correspond to little pixels ROI? in deep architecture this kind of seed will represent only few pixels? So how resolve the little seed size problem in this kind of image?

Figure 13: this type of graphic representation is not at all adapted to the presentation of the results. in fact, you draw lines linking the different species of seeds, but there is no link between the species. it is necessary to change this representation

In results, you never show image detection results? why?

in table 5 you present extremely close results and i do not see, for my part, if your proposal presents a statistically significant improvement. Moreover, you do not compare your results to other approaches in the bibliography. The same problem is found in Table 6.

best regards

Reviewer 3 Report

This article presents a system for classifying invasive plant seeds in real-time.

The article is well written, and the results support the conclusion.

However, I think that the authors should improve the quality of the manuscript because it presents several significant issues.

1. The introduction should provide a profound overview of the study.
And, in a certain way, it does up to line 59.
Then, the authors decided to introduce part for the description of
computer vision methods. In general, I think it can work as is, but the detection algorithms' description is too synthetic.
For example, in this part, the authors should clarify the meaning of
"one-stage" and "two-stage" in this context. Again, what do R-CNN, SSD, YOLO mean?
Please clarify this point to give a broader view of the chosen method.

2. Still in the introduction, "YOLOv5 has smaller weight file and faster reasoning speed of the model." needs at least a citation or something to support it.

3. Regarding the overview of the study, it is unclear what unique challenges
are associated with the task faced by the authors.
Therefore, the introduction should contain more details about the open research problems and
clarify the contributions of this work on how to address these research challenges.
Perhaps it could be better to separate the related work from the introduction.

4. In the introduction, the authors cite some works, even though
some important works with crucial advancements in the field are missing.
A non-exhaustive list is the following:
1) Using Deep Convolutional Neural Network for oak acorn viability recognition based on color images of their sections (https://www.sciencedirect.com/science/article/pii/S0168169918313930?via%3Dihub#!)
2) A novel deep learning based approach for seed image classification and retrieval (https://www.sciencedirect.com/science/article/pii/S0168169921002866)
3) On the Efficacy of Handcrafted and Deep Features for Seed Image Classification (https://www.mdpi.com/2313-433X/7/9/171)
4) An effective and friendly tool for seed image analysis (https://link.springer.com/article/10.1007%2Fs00371-021-02333-w)
5) A Convolution Neural Network-Based Seed Classification System (https://www.mdpi.com/2073-8994/12/12/2018)

5. Is there a specific purpose of separating 2.1 and 3.1 sections?
Perhaps they could be organized as a unique section (3.1).
Moreover, it is important to cite how the authors select the invasive plant seeds.
Also: is (or will be) the dataset publicly available?

6. Figure 12 caption should specify if the numbers are referred to the number of images or seeds for clarity.

7. Yolov5s is cited for the first time in Section 3.3 even though only Yolov5 has been cited up to this point. The authors should clarify why they adopted the small version of the detector.

8. Also, in 3.3, how did the authors choose the parameters?

9. In section 3.4, the speculation on the poor F1-score of Sida spinosa is reasonable
but could be detailed more. Specifically, for example, there is only a difference
of 100 elements between Sida Spinosa and Leucaena. Perhaps the reasons could also be related to other aspects?
Again, the results for Ipomoea and Sesbania also seem relatively low in respect to the others. Please give some motivation.
Finally, the authors should try some undersampling or
oversampling strategies to avoid this issue.

10. I suggest adding a figure representing all the seed types.
It could help the reader to understand the visual differences among them.

11. Figure 13 mentions "13 invasive plants". It should be 12.

12. In table 6, how did the authors conduct the experiments?
For example, why has ADAM been chosen in 5 out of 6 tests and SGD only once?
Please explain. Also, the authors should motivate the reasons why Exp4 obtained
such a poor performance.

13. In the conclusion section, the authors should emphasize more the real advantages
of the methods found and give some general points and suggestions to the authors
which take into account these fields.

Best regards.

Reviewer 4 Report

In this manuscript, Yang et al. applying deep learning model to plant seed detection and identification.  The author produced a plant seed image dataset, and proposed a YOLOv5 based attention detection model with efficiently detected seeds, which is a very interesting topic in plant phenomics study and had potential value for application. But with mistake in math formula deduce in main text, my suggestion is reconsider after these major revisions:
1.    The formula (5) in line is wrong in Line 180. And what x represent for was not clear described in formula (1-5). The left SiLU in Line 180 must be SiLU’.
2.    In Figure 13, a prediction accuracy is not enough, a training error also need measure to evaluate the performance of different models.  
3.    In Line 144-145, the value is should have a text description, that other can understand what these values standard for. 
4.    The writing and description of the method part were really terrible, it very had to get what the author did. Need re-write and origination. 
5.    Line 213, what |C\(A U B)| represent for?  
6.    In the Experiment part, it too short, the detail parameters must give in here.
7.    In formula 10,13, usually, a star * was used multiply, not x.
8.    Short abbreviation need define before use, e.g. line 198: FPN, PAN; Line 210 IOU, IoU, GIOU.

Round 2

Reviewer 3 Report

The authors responded to all my concerns.
I recommend acceptance of the manuscript.

Reviewer 4 Report

I am happy with the authors response, which answered my questions and updated in the manuscript. The manuscript can be acceptable with moderate English language update.